# Platform adaptive trial of novel antivirals for early treatment of COVID-19 In the community (PANORAMIC): protocol for a randomised, controlled, open-label, adaptive platform trial of community novel antiviral treatment of COVID-19 in people at increased risk of more severe disease

OG, EO and JA are joint first authors.
RH, PL and CCB are joint senior authors.

For numbered affiliations see end of article.

**Correspondence to**
Professor Christopher C Butler;
christopher.butler@phc.ox.ac.uk

Oghenekome Gbinigie [iD],[1] Emma Ogburn,[1] Julie Allen,[1] Jienchi Dorward [iD] ,[1,2] Melissa Dobson,[3] Tracie-Ann Madden,[4] Ly-Mee Yu,[1] David M Lowe,[5] Najib Rahman,[3,6,7] Stavros Petrou,[1] Duncan Richards,[8] Kerenza Hood,[4] Mahendra Patel,[1] Benjamin R Saville,[9,10] Joe Marion,[9] Jane Holmes,[1] May Ee Png [iD] ,[1] Gail Hayward [iD] ,[1] Mark Lown [iD] ,[11] Victoria Harris,[1] Bhautesh Jani,[12] Nigel Hart,[13] Saye Khoo,[14] Heather Rutter,[1] Jem Chalk,[1] Joseph F Standing,[15,16] Judith Breuer,[15] Layla Lavallee,[1] Elizabeth Hadley,[1] Lucy Cureton,[1] Magdalena Benysek,[1] Monique I Andersson [iD] ,[17] Nick Francis [iD] ,[11] Nicholas P B Thomas,[18,19,20] Philip Evans,[21,22] Oliver van Hecke,[1] Mona Koshkouei,[1] Maria Coates,[1] Sarah Barrett,[1] Clare Bateman,[1] Jennifer Davies,[1] Ivy Raymundo-Wood,[1] Andrew Ustianowski,[23] Jonathan Nguyen-Van-Tam,[24] Andrew Carson-Stevens,[4] Richard Hobbs,[1] Paul Little [iD] ,[11] Christopher C Butler [iD] [1]

## ABSTRACT

**Introduction** There is an urgent need to determine the safety, effectiveness and cost-effectiveness of novel antiviral treatments for COVID-19 in vaccinated patients in the community at increased risk of morbidity and mortality from COVID-19.

**Methods and analysis** PANORAMIC is a UK-wide, open-label, prospective, adaptive, multiarm platform, randomised clinical trial that evaluates antiviral treatments for COVID-19 in the community. A master protocol governs the addition of new antiviral treatments as they become available, and the introduction and cessation of existing interventions via interim analyses. The first two interventions to be evaluated are molnupiravir (Lagevrio) and nirmatrelvir/ritonavir (Paxlovid). Eligibility criteria: community-dwelling within 5 days of onset of symptomatic COVID-19 (confirmed by PCR or lateral flow test), and either (1) aged 50 years and over, or (2) aged 18–49 years with qualifying comorbidities. Registration occurs via the trial website and by telephone. Recruitment occurs remotely through the central trial team, or in person through clinical sites. Participants are randomised to receive either usual care or a trial drug plus usual care. Outcomes are collected via a participant-completed daily electronic symptom diary for 28 days post randomisation. Participants and/or their Trial Partner are contacted by the research team after days 7, 14 and 28 if the diary is not completed, or if the participant is unable to access the diary. The primary efficacy endpoint is all-cause, non-elective hospitalisation and/or death within 28 days of randomisation. Multiple prespecified interim analyses allow interventions to be stopped for futility or superiority based on prespecified decision criteria. A prospective economic evaluation is embedded within the trial.

**Ethics and dissemination** Ethical approval granted by South Central–Berkshire REC number: 21/SC/0393; IRAS project ID: 1004274. Results will be presented to policymakers and at conferences, and published in peer-reviewed journals.

**Trial registration number** ISRCTN30448031; EudraCT number: 2021-005748-31.

**STRENGTHS AND LIMITATIONS OF THIS STUDY**

⇒ PANORAMIC is a platform trial, enabling interventions to be added as the trial progresses, with interim analyses allowing interventions to be dropped as soon as prespecified criteria for superiority or futility are met, or for safety concerns.

⇒ In addition to recruitment by investigators at research sites, the research can be delivered 'direct-to-patient' through recruitment by a centralised team, with remote consent, follow-up and delivery of study medication to participants' homes, thereby not limiting participation to where people live or receive their healthcare.

⇒ A national inclusion and diversity strategy has been employed to actively promote the trial across the four UK nations to diverse communities and people from all backgrounds collaborating with the National Institute of Health and Care Research Clinical Research Network and equivalent networks in UK devolved administrations.

⇒ The open-label design means that it is not possible to quantify the contribution of any placebo-effect to treatment effects, but is more closely reflective of real-world practice.

## INTRODUCTION

The development and roll-out of national coronavirus disease 2019 (COVID-19) vaccination schemes has been transformative in reducing disease severity and to a lesser extent SARS-CoV-2 transmission.[1–3] Despite this, the emergence of new variants and waning immunity have led to intermittent surges in COVID-19 cases and hospitalisations.[4] The implementation of effective COVID-19 treatments therefore remains a critical management strategy and may be of great importance if future vaccine-escaping variants emerge. A number of drugs have been trialled as repurposed COVID-19 community treatments with evidence that some should not be used for this indication[5 6] while others are likely to be beneficial.[7 8] Directly-acting antiviral drugs are an important therapeutic approach, but evidence is limited.

Two new antiviral options are molnupiravir (Lagevrio) and nirmatrelvir/ritonavir (Paxlovid), with others being developed. Molnupiravir is a prodrug; the ribonucleoside analogue β-d-N4 -hydroxycytidine (NHC) is metabolised to NHC-triphosphate in cells, which when integrated introduces catastrophic hypermutation.[9] Paxlovid is a combination of nirmatrelvir and ritonavir; nirmatrelvir inhibits the activity of the SARS-CoV-2 3CL protease that is necessary for viral replication,[10] and ritonavir significantly slows the clearance of nirmatrelvir.[11]

Initial trials of molnupiravir and nirmatrelvir/ritonavir for COVID-19 have demonstrated safety and efficacy.[12 13] However, these trials were in unvaccinated patients prior to the omicron SARS-CoV-2 variant becoming prevalent, and it is not clear if there are particular subgroups of patients who should be prioritised for treatment. Furthermore, the impact on viral load, antiviral resistance and emergence of new variants requires further evaluation, and cost-effectiveness of these agents at scale is as yet unknown. The impact on long COVID is also yet to be assessed. Nevertheless, these encouraging efficacy trials, and the likelihood that further plausible treatments will become available and require evaluation, justifies a large-scale, ongoing, pragmatic evaluation of antiviral treatments for use in the community in a largely vaccinated population, while current variants are circulating, to rapidly generate robust evidence for guiding decisions about widespread deployment.

We therefore established an adaptive multiarm platform trial with a master protocol to test whether novel antiviral agents are safe, effective and cost-effective treatments for people in the community with COVID-19 who are at increased risk of an adverse outcome.

### Objective

To assess the effectiveness and cost-effectiveness of novel antiviral treatments in reducing all-cause, non-elective hospitalisation and/or death within 28 days of randomisation among patients with test-positive COVID-19 in the community and who are at increased risk of requiring hospital treatment.

## METHODS AND ANALYSIS
### Trial design

The Platform Adaptive trial of NOvel antiviRals for eArly treatMent of COVID-19 In the Community (PANORAMIC) is an open-label, prospective, adaptive platform, randomised clinical trial in community care.

A multiarm 'platform trial' is a clinical trial that allows for multiple treatments for the same disease to be tested simultaneously under a single master protocol. Prespecified adaptations allow interventions to be added to the trial, or stopped for futility or superiority while the trial is in progress through prespecified interim analyses.[14 15] Participants are randomly assigned to either usual care, or usual care plus a trial intervention. Usual care represents the standard care that participants would receive via the National Health Service (NHS), and is largely supportive, apart from for those at the highest risk of an adverse outcome.[16]

The master protocol defines a priori decision rules to allow for dropping a treatment for futility or declaring a treatment superior to usual care.[17] If at an interim analysis, usual care plus an antiviral is deemed superior to usual care alone for the primary endpoint of all-cause, non-elective hospitalisation and/or death within 28 days of randomisation, the superior treatment may be incorporated into usual care as the new standard of care. Cost-effectiveness will also be assessed. A subset of participants is additionally enrolled into a virology substudy, and are asked to provide nasopharyngeal swabs and fingerpick blood samples at intervals over the 14 days following recruitment.

The first and second antivirals to be evaluated in PANORAMIC are molnupiravir[18] and nirmatrelvir/ritonavir, respectively.

### Patient and public involvement

Patient and public involvement (PPI) contributors contribute to refining the study question, design,

implementation, interpretation and dissemination of findings. At trial conception, the aims and design of the study were discussed with members of the public who had experience of COVID-19, either personally or through household members, and who were at higher risk of complications from COVID-19. PPI groups supporting the trial include an ethnically diverse main study PPI group who have advised on patient facing documents and study processes, and have helped to draft easy read versions of study documents. In addition, bespoke PPI groups established in Northern Ireland, Scotland and Wales have advised on data capture and recruitment processes specific to their local health systems, and will contribute to advise on dissemination. Two PPI contributors sit on the Trial Steering Committee to help guide trial progress. A co-investigator has a specific remit for community engagement, developing and implanting initiatives with the support of pharmacy networks to ensure uptake especially in areas of higher social deprivation and among minority ethnic groups: feedback about all aspects of the trial is received from this community engagement programme.

## Study setting

The trial is implemented by the University of Oxford Primary Care and Vaccines Collaborative Clinical Trials Unit (PCV-CTU)[19] with further support from the Oxford Respiratory Trials Unit and the Centre for Trials Research, Cardiff University, supported by the National Institute of Health and Care Research Clinical Research Network, the National Institute of Health and Care Research and the Department of Health and Social Care (and equivalents in devolved administrations).

The PCV-CTU is able to act as a central recruiting site, and PANORAMIC Hubs act as clinical recruitment sites. PANORAMIC Hubs are clinical sites that include General Practice (GP) sites as single practices or a federation of practices that are able to operate under a single site agreement with a principal investigator to undertake study procedures as detailed in the master protocol. Hubs can include GP practices, community trusts and other healthcare providers. Potential participants can be referred to Hubs by other healthcare facilities for screening. As well as recruiting patients through routine consultations, Hubs perform database searches for COVID-19 positive test results in registered patients who are clinically vulnerable (see box 1), and invite them to take part in the trial. All mandated study procedures can be conducted remotely, in keeping with the prevailing self-isolation advisory governmental guidance for patients with COVID-19 in the community.[20]

## Eligibility criteria

The inclusion criteria are: patient or their legal representative is able and willing to provide informed consent; patient presenting with symptoms attributable to COVID-19 starting within the past 5 days and ongoing; patient has a positive SARS-CoV-2 test (lateral flow test

---

**Box 1  Criteria considered to make a potential participant at higher risk of worse outcomes from COVID-19**

⇒ Chronic respiratory disease (including chronic obstructive pulmonary disease, cystic fibrosis and asthma requiring at least daily use of preventative and/or reliever medication).
⇒ Chronic heart or vascular disease.
⇒ Chronic kidney disease.
⇒ Chronic liver disease.
⇒ Chronic neurological disease (including dementia, stroke, epilepsy).
⇒ Severe and profound learning disability.
⇒ Down's syndrome.
⇒ Diabetes mellitus (type 1 or type 2).
⇒ Immunosuppression: primary (eg, inherited immune disorders resulting from genetic mutations, usually present at birth and diagnosed in childhood) or secondary due to disease or treatment (eg, sickle cell, HIV, cancer, chemotherapy).
⇒ Solid organ, bone marrow and stem cell transplant recipients.
⇒ Morbid obesity (body mass index >35 kg/m²).
⇒ Severe mental illness.
⇒ Care home resident.
⇒ Judged by recruiting medically qualified professional, research nurse, nurse prescriber, prescribing pharmacist, dependent on the Intervention Specific Appendix for the specific antiviral involved, to be clinically vulnerable.

---

and/or PCR) up to 2 days before symptom onset and randomisation; and, patient is aged ≥50 years or aged 18–49 years with an underlying chronic health condition considered to make them clinically vulnerable (see box 1). Exclusion criteria are: patient currently admitted to hospital (inpatient); patient previously randomised in the PANORAMIC trial; and, patient currently participating in a clinical trial of a therapeutic agent for acute COVID-19. Additional exclusion criteria specific to each intervention arm, if any, are listed in the Intervention Specific Appendices (ISAs) of trial arms within the master protocol. Patients must be eligible for at least two arms (usual care and at least one novel antiviral intervention).

## Study procedures
### Recruitment

The entire recruitment process can be done remotely as well as in person. Potential participants can register via the trial website, through a free-phone telephone call to the central trial team, or via a PANORAMIC Hub.

### Informed consent, screening and enrolment

Eligibility is assessed at a PANORAMIC Hub, other NHS healthcare provider or by the central clinical trial team, by a suitably trained and experienced medically qualified professional, research nurse, nurse prescriber or prescribing pharmacist, as determined by the regulator and specified in the ISA for the specific antiviral involved.

Prospective participants are provided with written, pictorial and/or verbal versions of the Patient Information Sheet, detailing the nature of the trial and the known side-effects/risks involved in taking part. Participants provide consent to participate through a two-way

discussion (apart from those who lack capacity to do this) either face-to-face or by a telephone/video call. Prospective participants with capacity and being recruited in-person provide written informed consent (see online supplemental additional file 1). Consent forms for participants recruited in-person via PANORAMIC Hubs are filed in participants' medical notes, with a printed copy given to the participant. Participants recruited remotely provide consent using an electronic consent form that is held securely on the trial database. Participants can either download their consent form, or a hard copy of the consent form is posted to them.

Prospective participants lacking capacity to consent are only eligible if they live in a care home. If the recruiting healthcare professional considers that a patient in a care home lacks capacity to provide consent for themselves, then a personal or professional legal representative (England and Wales only) is asked to provide consent in-person or remotely.

Participants who are unable or too unwell to complete baseline information or respond to surveys for themselves can identify a Trial Partner to assist them in: completing the initial screening questionnaire and baseline information; completing the informed consent forms; and, completing the electronic symptom diary (see 'follow-up' section). A letter is issued to Trial Partners, informing them of the study and notifying them that they have been nominated for this role by the prospective participant.

### Randomisation and blinding

Participants are randomised using a secure, fully validated and compliant web-based randomisation system embedded within Spinnaker (a data entry system), with binary stratification by age (<50 years vs ≥50 years) and vaccination status (yes vs no). Participants are randomised to one trial arm using fixed equal allocation ratios corresponding to the number of eligible arms in the trial. For example, if there are two active interventions (A and B), the allocation ratio will be 1:1:1 for usual care, active A, active B (respectively), such that 33% of participants are randomised to usual care. If there are three active interventions, the allocation ratio will be 1:1:1:1, such that 25% of participants are randomised to usual care. As this is a nationwide, individually randomised trial that aims to include large numbers of participants, individual participant characteristics and infecting strain types of SARS-CoV-2 are expected to be equally distributed between trial arms.

PANORAMIC is an open-label trial. The participant, legal representative (if applicable) and recruiting clinician know the participant's allocation. The trial team and recruiting clinicians are kept blind to emerging results of interim analyses. Only the unblinded statisticians and the independent members of the Data and Safety Monitoring Committee have access to unblinded interim results corresponding to a given intervention, until such time as a decision is made to close the intervention.

### Baseline assessments

During screening and enrolment, participants and/or recruiting clinicians record baseline data including: date of birth; sex; ethnicity; vaccination status; qualifying comorbidities; symptoms and severity; a measure of their health-related quality of life (EuroQoL EQ-5D-5L)[21]; number of household contacts; and, whether they have received a monoclonal antibody treatment for COVID-19.

### Interventions

PANORAMIC trial is testing novel antiviral agents that have the potential to be widely used to treat COVID-19 in the community. Each agent is fully specified in an ISA. The antiviral drugs are couriered to participants, typically within 24 hours of randomisation. Pharmacies can supply antivirals to participants via community pharmacy services or online pharmacy services. The antivirals can also be collected from a pharmacy by the participant (or someone on their behalf, with appropriate infection control measures).

PANORAMIC is a randomised controlled, open-label, pragmatic trial.[22 23] The control arm is usual care. Usual care can include antiviral treatment available to individual patients in routine care in the NHS.[24] In the UK, patients at highest risk are able to access antiviral treatments directly from the NHS via COVID Medicine Delivery Units and analogous organisations; otherwise, in the absence of complicated infection (eg, bacterial superinfection), Usual care in the NHS is generally supportive.[24] Participants assigned to an intervention arm additionally receive the usual care through the NHS that they would ordinarily have received, had they not participated in the trial. The trial team are not involved in making clinical or clinical management decisions for participants. Participants receiving a monoclonal antibody infusion or an antiviral agent as part of their usual care were eligible to receive a (different) antiviral through the trial. However, those at highest risk of an adverse outcome were informed that they were eligible for access to antiviral treatment through NHS services.

### Follow-up

Following randomisation, participants in the intervention arm receive a participant pack containing: the allocated antiviral agent; an information booklet; a participant card detailing how the medication should be administered, precautions and safety guidance; a medication appendix providing further information about the allocated intervention; an emergency card with a phone number with a 24-hour phone line to access an on-call clinician for safety concerns; and, a pregnancy test to be used by participants of childbearing potential for certain interventions.

All participants are emailed a link each day to an online symptom diary and are asked to complete it daily for 28 days. Participants are asked: to rate a variety of symptoms (such as fever, cough, breathlessness and fatigue) on an ordinal scale (eg, 'no problem,' 'mild problem,' 'moderate problem' or 'major problem'); whether they

have been hospitalised or required contact with health and social services; how they are feeling on a scale of 0–10 (0 being the worst one can imagine and 10 being the best one can imagine); whether they feel fully recovered; whether they are taking over-the-counter medication; whether the number of people in the household has changed; confirm whether they have taken the antiviral agent (if applicable); and, at fortnightly intervals the EQ-5D-5L to assess their health-related quality of life. The central trial team calls participants/Trial Partners with no internet access and those who have not completed their diary for at least two consecutive days before days 7, 14 and 28.

All participants will receive a phone call from the trial team on Day 2 of the trial to confirm receipt of trial materials, confirm consent and understanding of follow-up procedures and to answer any queries. Participants receiving an antiviral agent receive additional safety calls from members of the trial team, to determine whether participants are experiencing adverse effects, and, if applicable, to ensure that participants who are physiologically capable of becoming pregnant and who are not using highly effective contraception confirm a negative pregnancy test result prior to starting the intervention. The exact schedule of safety calls is intervention-dependent, and outlined in each ISA.

To investigate the impact of trial interventions on the longer-term effects of COVID-19, we contact participants at 3 and 6 months after randomisation to ascertain well-being, persistence of symptoms perceived to be related to the index COVID-19 illness, and longer-term consequences. Participants' medical record data may additionally be accessed up to 12 months following enrolment to gather follow-up data from enrolment to 6 months. Sources of routinely collected data (eg, NHS Digital) may also be used to follow-up participants for up to 10 years.

### Study outcomes
The primary endpoint is all-cause, non-elective hospitalisation and/or death within 28 days of randomisation, ascertained through patient/Trial Partner report, and/or patient medical records. Secondary endpoints include: time to self-reported recovery defined as the first instance that a participant report feeling fully recovered from the illness; duration of symptoms; symptom recurrence; daily rating of feeling well reported by participants; healthcare service use; participant reported new COVID-19 infections in their household; safety and cost-effectiveness outcomes; symptoms; and, well-being at 3 and 6 months (with determination of proportion reporting symptoms perceived to be related to long COVID) from randomisation.

### Data collection and management
Data are entered into electronic case report forms (CRFs) by the participant, their Trial Partner or a Hub team member, using Spinnaker. Spinnaker is an online secure, Food and Drug Administration part 11B compliant, data entry system, which is designed to collect sensitive data,

such as participant and Trial Partner contact details. All identifiable participant data are encrypted using the Advanced Encryption Standard. Data are stored on a secure cloud hosted server physically located in London, UK. Participant and Trial Partner data will be kept and stored securely for as long as required by the trial and reviewed on an annual basis.

### Statistical methods
*Primary endpoint analysis*
Details of the statistical design and methods are described in a Master Statistical Analysis Plan and Adaptive Design Report (ADR). The primary endpoint analysis is a Bayesian logistic regression model of the primary endpoint comparing a given intervention versus usual care, adjusting for age, comorbidity status and vaccination status.

The trial design incorporates multiple prespecified interim analyses that allow each intervention to stop early for futility or superiority. If the Bayesian posterior probability of beneficial treatment effect (alternative hypothesis) is greater than or equal to a prespecified threshold at an interim or final analysis, the null hypothesis (no beneficial intervention effect) is rejected, and the intervention is deemed superior to usual care with respect to hospitalisation/death. The decision criteria are defined in the ADR and control the Type I error at the traditional 0.05 two-sided level for each intervention, accounting for multiple interim analyses. As described in the ADR, the prespecified interim analyses may be bypassed for a given intervention at the discretion of the blinded Trial Management Group (TMG) in the event of a fast accrual rate. The success thresholds at final and interim analysis are prespecified and dependent on the number of interim analyses, which is a function of the speed of enrolment. The ADR also contains extensive simulations to explore the performance of the adaptive design, including power and Type I error. Subgroup analyses are performed according to age group, baseline comorbidity status, severity of symptoms at baseline, duration of symptoms at baseline, use of an inhaled corticosteroid steroid at randomisation or during 28 days of follow-up, swab positivity status (PCR positive vs Lateral Flow Device positive), vaccination status and COVID-19 risk category (as per the UK government description). Details of subgroup analyses can be found in the statistical analysis plan. All statistical analyses of primary and some secondary outcome data analyses are performed by Berry Consultants and the University of Oxford. Berry Consultants is based in the USA; as such they will not receive identifiable trial data.

### Sample size
The master protocol specifies a maximum sample of approximately 5300 participants per arm, which provides approximately 90% power for detecting a 33% relative reduction in the risk of hospitalisation/death in an experimental arm relative to usual care, based on the assumption of an underlying 3% hospitalisation/death rate in the

usual care arm, and an intervention lowering the hospitalisation/death rate to 2%. However, an intervention-specific appendix may define an alternative maximum sample size for an intervention based on alternative assumptions for a given intervention or based on the relevant state of the pandemic. For example, if the severity of COVID-19 weakens and the aggregate (blinded) primary endpoint event rate is lower than expected, the maximum sample size may be increased to ensure sufficient statistical power.

### Primary analysis population

For each intervention, the primary analysis population includes all concurrently randomised patients who were eligible to be randomised to an antiviral agent (concurrent and eligible), who fulfil the eligibility criteria and who have had the opportunity to complete 28 days of follow-up. Eligible participants will be analysed according to the group they were randomised to regardless of deviation from protocol.

### Safety monitoring

Symptoms, potential medication side-effects and serious adverse events (SAEs) are collected from participant daily diaries, calls to participants/Trial Partners, face-to-face visits with Hub clinicians, medical records, notes reviews and data extracts from hospital and primary care medical records from all UK devolved administrations.

A risk assessed and proportionate approach to safety monitoring is adopted for each antiviral included in the trial. In line with the Summary of Product Characteristics or Investigator Brochure, the risks and the safety profile for each antiviral agent are assessed, and the mitigation and monitoring procedures are detailed in the ISA. All safety procedures will be according to University of Oxford Primary Care Clinical Trials Unit pharmacovigilance Standard Operating Procedures.

For each antiviral agent, we only collect adverse events (AEs), other than those prespecified symptoms collected via the participant diaries, if and when specified in the relevant ISA. For certain interventions, pregnancy occurring within 28 days of first intervention administration is recorded as an AE of Special Interest. All-cause hospitalisation and/or death is the primary outcome, and these data are captured in CRFs. SAEs other than hospitalisation or death due to COVID-19 are reported for all antiviral agents over the follow-up period. Hospitalisations for pre-existing conditions, including elective procedures planned prior to trial entry, which has not worsened, do not contribute to our primary outcome, and do not constitute SAEs.

A risk assessment and monitoring plan is prepared before opening recruitment to each antiviral agent and is reviewed as necessary over the course of the trial to reflect significant changes to the protocol or outcomes of monitoring activities. Monitoring is performed by the Primary Care Clinical Trials Unit. The level of monitoring required is informed by the risk assessment.

### Virology substudy

A subset of patients from the intervention and comparator arms of the trial are invited to participate in a virology sampled cohort for additional virological testing. The primary aim of the virology sampled cohort is to determine whether the antiviral treatment under study reduces viral load to undetectable levels sooner, and to explore the effect of antiviral treatment on development of antiviral resistance. The substudy primary outcome is SARS-CoV-2 viral load at Day 7. Secondary outcomes include SARS-CoV-2 viral load Days 0–7 and Day 14; SARS-CoV-2 viral genetic whole genome sequence at Day 1, Day 5 (±1 day) and Day 14 (±1 day) and SARS-CoV-2 antibodies at Day 1, Day 5 (±1 day) and Day 14 (±1 day); and to identify any common viral genetic mutations (occurring in greater than 1% of patients) in patients receiving novel antiviral(s).

The eligibility criteria are as for participants taking part in the main trial, but with an additional exclusion criterion: participants who are within 3 months of receiving a non-trial anti-SARS-CoV-2 antibody therapy are ineligible.

Up to approximately 300 participants from each trial intervention arm and the usual care arm are recruited into the voluntary virology sampled cohort. The first 30 patients enrolling from each trial arm undergo intensive daily viral load monitoring, and are asked to provide daily nasopharyngeal self-taken swabs for 7 days, and an additional nasopharyngeal swab on Day 14 (±1 day). For participants in intervention arms, the first sample will be taken immediately prior to commencing antiviral treatment (Day 1). The remaining 270 from each arm in the virology samples cohort have less intensive viral load monitoring, and are asked to provide three nasopharyngeal swabs: one prior to starting treatment, one on Day 5 (±1 day) and one on Day 14 (±1 day).

All participants are asked to take three fingerprick dried blood spot samples: one pretreatment, one on Day 5 (±1 day) and one on Day 14 (±1 day). Participants consenting to take part in the virology sampled cohort are sent CE-IVD approved (that is, compliant with the European In-Vitro Diagnostic Devices Directive) sampling kits for nasopharyngeal sampling, dried blood spot sampling, prepaid postage and packaging, to post samples to the virology processing site. Samples taken at home should be posted to the trial team within 3 days of sampling, and ideally within 24 hours.

### Health economic evaluation

A prospective economic evaluation is embedded within the trial design to assess the cost-effectiveness of each antiviral from an NHS and Personal Social Services (PSS) perspective. The resource inputs associated with embedding each trial antiviral treatment into routine clinical practice are estimated. Broader resource use is drawn from linked routine health data—encompassing primary care encounters, hospital inpatient/day case admissions, outpatient visits and accident and emergency attendances.

Unit costs are valued using national reference tariffs and attached to resource inputs to generate a compound total NHS and PSS cost per trial participant over the trial time horizon. EQ-5D-5L data are converted using standard algorithms into utility scores for quality-adjusted life year (QALY) estimation. Cost-effectiveness is expressed as incremental cost per QALY gained.[25] Secondary expressions of cost-effectiveness include incremental cost per hospitalisation and/or death prevented over 28 days. Bivariate regression of costs and measures of health consequence, with multiple imputation of missing data, will be conducted to generate within-trial estimates of incremental cost-effectiveness. Sensitivity analyses will assess the impact of areas of uncertainty surrounding components of the economic evaluation. If economic outcomes are non-convergent within the trial follow-up period, then extrapolation of cost-effectiveness through decision-analytical modelling will be considered, drawing on the best available information from the literature to supplement the trial data. Specific plans for the economic evaluation are outlined in a prespecified health economics analysis plan.

## ETHICS, APPROVALS, MONITORING AND DISSEMINATION

The trial has been approved by the University of Oxford Research Governance Ethics and Assurance Team as study sponsor, the South Central–Berkshire Research Ethics Committee (REC number: 21/SC/0393) of the Health Research Authority, and the UK Medicines and Healthcare products Regulatory Agency. All participants provide informed consent, online or by telephone, before participation. All participants completing the 28-day follow-up are provided with a £10 voucher in recognition of their contribution to the study.

The University of Oxford as sponsor has a specialist insurance policy in place, which would operate in the event of any participant suffering harm as a result of their involvement in the research (Newline Underwriting Management, at Lloyd's of London). NHS indemnity operates in respect of the clinical treatment that is provided.

An independent Data Monitoring and Safety Committee reviews emerging data provided by the Statistical Analysis Committee and communicates key decisions to the Trial Steering Committee, which in turn advises the TMG and also provides trial oversight.

It is expected that trial results will be published in peer-reviewed journals and relevant findings presented at national and international conferences.

## Trial status

PANORAMIC was registered on the ISRCTN registry on 3 November 2021. Enrolment started on 8 December 2021. By 17 September 2022, 26 285 participants had been recruited. Protocol V.5.0, 9 May 2022 (see online supplemental additional file 2).

## DISCUSSION

### Summary

Despite the high uptake of vaccination against COVID-19 in many countries, the disease remains prevalent, with many patients continuing to experience considerable morbidity and require treatment in hospital. We describe a platform randomised trial to evaluate antiviral therapeutic agents for use by people at higher risk from COVID-19 in the community with confirmed acute, symptomatic SARS-CoV-2 infection.

### Comparison with other studies of novel antiviral agents for community treatment of COVID-19

A phase 3 placebo-controlled, randomised trial of molnupiravir recruited 1433 COVID-19 outpatients in over 20 countries, with a primary efficacy endpoint of all-cause hospitalisation or death within 29 days of enrolment (Molnupiravir for Oral Treatment of Covid-19 in Nonhospitalized Patients - 'MOVe-OUT' trial).[26] The authors found that treatment with molnupiravir reduced the risk of hospitalisation or death compared with placebo (risk difference, −3.0%; 95% CI: −5.9% to −0.1%).[26] Adverse events occurred with similar frequency in molnupiravir and placebo groups (30.4% and 33.0 %, respectively), as did adverse events deemed to be related to the trial regimen (8.0% and 8.4%, respectively). No deaths were attributed to the trial treatment (one death in the molnupiravir group and nine deaths in the placebo group).

As in the PANORAMIC trial, participants were at higher risk of an adverse illness course, received a 5-day course of molnupiravir at a dose of 800 mg two times per day, and received the intervention within 5 days of symptom onset. However, the trial recruited unvaccinated patients; the vast majority of the UK adult population are multiply-vaccinated (primary course plus boosters).[27] Furthermore, Delta, Gamma and Mu variants accounted for the majority of SARS-CoV-2 variants in the MOVe-OUT trial,[28] whereas the predominant variant in circulation in the UK has been Omicron since December 2021.[29] PANORAMIC additionally incorporates an assessment of the impact of antiviral drugs on viral load and markers of viral resistance.

In a phase 2–3 randomised, placebo-controlled trial of 2246 outpatients with COVID-19 from the USA (41%), Europe (30%), South America (12.3%), Asia (14%) and Africa (0.6%), at higher risk of an adverse illness course, treatment with nirmatrelvir/ritonavir resulted in a 5.8% absolute risk reduction in the primary outcome of COVID-19 related hospitalisation and all-cause death within 28 days (0.72% and 6.53%, respectively, risk difference −5.81%, 95% CI: −7.78% to −3.84%, p<0.0001).[13] Viral load was significantly reduced by treatment with nirmatrelvir/ritonavir (adjusted mean difference of

−0.868 log10 copies per millilitre, 95% CI: −1.074 to −0.6615, p<0.001). The incidence of adverse events was similar in both groups, and all 13 deaths occurred in the placebo group. The trial population was again unvaccinated, and therefore distinct from the UK population taking part in the PANORAMIC trial.

### Strengths and limitations

The platform design, informed by the experience of the Platform Randomised Trial of Treatments in the Community for Epidemic and Pandemic Illnesses (PRINCIPLE) trial,[30] allows PANORAMIC to add new interventions to the trial as they become available; this increases the efficiency of the trial as multiple interventions can be assessed by a single trial platform without having to set up a new trial each time a new intervention for this condition requires evaluation. Prespecified interim analyses allow randomisations to interventions to be stopped as soon as prespecified criteria for superiority or futility are met, potentially reducing time to trial conclusions. This ensures the trial's relevance in the face of rapidly evolving pandemic circumstances.

Deploying antimicrobials of any kind at scale raises the question of their possible impact on antimicrobial resistance. A virology substudy has been incorporated in PANORAMIC, which allows us to estimate virological endpoints, as well as facilitating careful evaluation of potential harms associated with antiviral treatment, such as the development of antiviral resistance and emergence of new variants.

Cost-effectiveness of novel antivirals is as yet unknown, but is critically important to considerations of widespread deployment of expensive: PANORAMIC aims to fill this gap in the evidence base for these agents.

Traditionally, primary care research implementation has followed a similar model to hospital-based studies, in which the 'participant comes to the research'. In this approach, potential participants are invited to participate if they receive their healthcare or live in the proximity to the research site. The capacity of PANORAMIC for recruitment of eligible people from almost anywhere in the UK, not limited by where people live or receive their healthcare, allows the 'research to be taken to the patient'. This is particularly important, given that participants are ill and probably highly infectious.

The trial has been designed to be minimally burdensome for participants; all trial procedures are possible remotely, from registration, to eligibility checks, to receiving trial medications and virology substudy materials by courier. This has facilitated rapid recruitment to the trial, with over 26 000 participants recruited to date. PANORAMIC strives to be a truly representative trial, with participants from various backgrounds recruited nationally from all four UK nations. A proactive outreach strategy has been employed, led by the trial's national pharmacy, and inclusion and diversity lead, with the support of UK-wide pharmacy networks, to help to promote the trial to diverse communities and to those disproportionately affected by COVID-19. This includes people from ethnic minority backgrounds and those living in areas of higher deprivation, traditionally known to be under-represented in clinical trials. The proportion of PANORAMIC participants in the molnupiravir versus usual care comparison older than 50 years who are from ethnic minorities is approximately 5%, which is not dissimilar to that in the English and Welsh general population (just over 6%).[31] However, we recognise that recruitment to the trial requires prospective participants to navigate the registration process, which might mean that people from certain groups, such as non-English speaking populations, may be less likely to enrol in the trial.

In addition to the primary outcome that is measured at 28 days, PANORAMIC evaluates longer-term outcomes at 3 and 6 months, which will help ascertain the effect of antiviral treatment on long COVID. Long COVID, defined as symptoms beyond 4 weeks after index illness[32] may affect between 10%[33] and 43.4%[34] of patients with COVID-19, and is characterised by a range of physical and psychological symptoms.[32] Thus far, we do not know whether novel antiviral treatments reduce symptoms associated with the acute illness over the longer-term.

Some may consider the open-label design of the trial a weakness. The lack of blinding means that we cannot estimate the proportion of any positive effect from the treatment that results from a possible placebo effect. Performance bias is more likely to affect outcomes that are considered subjective, such as symptom or wellness ratings. However, the objective primary outcome in PANORAMIC (non-elective hospitalisation and/or death) is unlikely to be affected by a placebo effect, as hospital admission is a clinical decision, and the virology substudy will also provide a helpful pointer as to whether the treatments are effective.[18] Furthermore, comparison with usual care is in keeping with pragmatic trial design and more closely reflective of real-world practice.[35] As placebos are not used in clinical care, the results of an open-label trial are more likely to reflect what would happen if the intervention were introduced into routine clinical practice,[35] additionally enabling a more realistic assessment of cost-effectiveness. Findings from the pragmatic, open-label PRINCIPLE trial have found no difference in outcome measures that rely on participants' self-reported recovery between participants allocated to usual care and usual care plus a study drug.[5 30 36]

**Author affiliations**

[1]Nuffield Department of Primary Care Health Sciences, University of Oxford, Oxford, UK

[2]Centre for the Aids Programme of Research in South Africa (CAPRISA), University of KwaZulu–Natal, Durban, South Africa

[3]Nuffield Department of Medicine, Oxford Respiratory Trials Unit, Oxford, UK

[4]Centre for Trials Research, Cardiff University, Cardiff, UK

[5]University College London, Institute of Immunity and Transplantation, London, UK

[6]NIHR Oxford Biomedical Research Centre, Oxford, UK

[7]Chinese Academy of Medicine Oxford Institute, University of Oxford, Oxford, UK

[8]Nuffield Department of Orthopaedics, Rheumatology and Musculoskeletal Sciences (NDORMS), University of Oxford, Oxford, UK

[9]Berry Consultants, Austin, Texas, USA

[10]Department of Biostatistics, Vanderbilt University Medical Center, Nashville, Tennessee, USA

[11]Primary Care Research Centre, University of Southampton, Southampton, UK

[12]General Practice and Primary Care, School of Health and Wellbeing, MVLS, University of Glasgow, Glasgow, UK

[13]School of Medicine, Dentistry and Biomedical Sciences - Centre for Public Health, Queen's University Belfast, Belfast, UK

[14]Department of Pharmacology, University of Liverpool, Liverpool, UK

[15]Infection Inflammation and Immunology, UCL Great Ormond Street Institute of Child Health Population Policy and Practice, London, UK

[16]Department of Pharmacy, Great Ormond Street Hospital for Children, London, UK

[17]Department of Microbiology, Oxford University Hospitals NHS Foundation Trust, Oxford, UK

[18]Windrush Medical Practice, Witney, UK

[19]Thames Valley and South Midlands Clinical Research Network, National Institute for Health and Care Research, Oxford, UK

[20]Royal College of General Practitioners, London, UK

[21]Faculty of Health and Life Sciences, University of Exeter, Exeter, UK

[22]National Institute for Health Research Clinical Research Network, London, UK

[23]Regional Infectious Diseases Unit, North Manchester General Hospital, Manchester, UK

[24]Lifespan and Population Health Unit, University of Nottingham School of Medicine, Nottingham, UK

**Acknowledgements** The authors would like to especially thank all participants in the study and acknowledge the work and support of all participating general practices, NHS COVID-19 treatment services and other health and social care organisations supporting the trial. We would also like to thank our PPI contributors, the Trial Steering Committee, the Data Monitoring and Safety Committee, primary care and other colleagues in the National Institute of Health and Care Research Clinical Research Network, and the National Institute of Health and Care Research. CCB acknowledges part support as senior investigator of the National Institute of Health and Care Research, the NIHR Community Healthcare Medtech and In-Vitro Diagnostics Co-operative (MIC), and the NIHR Health Protection Research Unit on Health Care Associated Infections and Antimicrobial Resistance. RH acknowledges his part-funding from the NIHR Applied Research Collaboration (ARC OxTV) and the NIHR Community Healthcare Medtech and In-Vitro Diagnostics Co-operative (MIC). GH is funded by and NIHR Advanced Fellowship and by the NIHR Community Healthcare Medtech and In-Vitro Diagnostics Co-operative (MIC). JD is funded by the Wellcome Trust PhD Programme for Primary Care Clinicians (216421/Z/19/Z). SP receives support as an NIHR Senior Investigator (NF-SI-0616-10103) and from the UK NIHR Applied Research Collaboration Oxford and Thames Valley. OvH is supported by an NIHR Development and Skills Enhancement Award. KH receives support as an HCRW Senior Research Leader and the Centre for Trials Research receives infrastructure funding from Health & Care Research Wales and Cancer Research UK. OG is funded by ECRAID-Prime (grant number 101046109). Data Monitoring and Safety Committee Independent members: Professor Deborah Ashby (Chair), Professor Benjamin Fisher, Professor Simon Gates, Professor Gordon Taylor, Professor Martin Underwood. Trial Steering Committee Independent members: Philip Hannaford (Chair), Corina Cheeks, Professor Ranjit Lall, Professor Alastair Hay, Professor William Hollingworth, Professor Matthew Sydes: Independent observer, Professor Mike Moore: Independent observer.

**Contributors** CCB and JN-V-T conceived the study. CCB is the chief investigator. PL and RH are co-chief investigators. CCB, PL and RH decided to publish the paper. BRS, L-MY, JH, MD, CCB, RH, PL, GH, OG, JD, NR, DR, SP, DML, JFS, KH, PE, OvH and ML provided input to the trial design. EO, JA, PE, LL, EH, LC, MB, MC, SB, CB, JD, AC-S and IR-W are responsible for study implementation and acquisition of data. CCB, OG, GH, RH, JH, L-MY, JD, JM, BRS, EO, JA, MP, PL, KH, NR, JFS and SP drafted the manuscript. HR leads the clinical team. L-MY, BRS, JH, VH and JM contribute to statistical analysis. SK, DR, NR and MD provide input to safety evaluations, monitoring and drug interactions. MP is the National Pharmacy, and Inclusion and Diversity Lead for the trial. SP and MEP run the economic evaluation. JFS, DML and JB lead the virology substudy. JC leads on the information systems. MB leads data management. CCB, PL, OG, NR, SP, DR, KH, MP, BRS, EO, JD, DML, SK, NF, NPBT, PE, JFS, JB, JA, MD, T-AM, MEP, GH, ML, BJ, NH, JC, EH, LC, MB, MIA, OvH, AU, MK, L-MY and RH are members of the Trial Management Group supporting site recruitment, activity and delivery. OG and CCB produced the first draft of the manuscript. All authors critically revised the manuscript. All authors are contributing to the conduct of the trial.

**Funding** This work is supported by the UKRI/NIHR, Grant number NIHR135366. Virology team members are partly supported by an MRC COVID modelling grant, Grant number MR/W015560/1. JD is funded by the Wellcome Trust PhD Programme for Primary Care Clinicians (216421/Z/19/Z). For the purpose of Open Access, the author has applied a CC BY public copyright license to any Author Accepted Manuscript version arising from this submission. NH was funded by the Northern Ireland Clinical Research Network, Public Health Agency. OG is funded by ECRAID-Prime (grant number 101046109).

**Competing interests** JN-V-T was seconded to the Department of Health and Social Care, England (DHSC) from October 2017 to March 2022. The views expressed in this paper are those of its authors and not necessarily those of DHSC. JN-V-T reports a lecture fee from Gilead Sciences Ltd (manufacturer of remdesivir) and a paid Influenza Advisory Board for F. Hoffmann-La Roche (manufacturer of tocilizumab), both after March 2022. KH is a member of the following NIHR committees: HTA General Committee, HTA Funding Strategy Group, Research Professors Funding Committee. KH is co-investigator on the grant provided by UKRI/NIHR, Grant number NIHR135366 (subcontract from University of Oxford to Cardiff University). KH received a grant from AstraZeneca to support a trial of Evusheld for the prevention of COVID in high-risk individuals (to Cardiff University). KH is an independent member of the IDMC for the OCTAVE-DUO trial of vaccines for COVID in high-risk individuals (unpaid). DML has received grants/contracts from LifeArc, Medical Research Council, Bristol Myers Squibb and Blood Cancer UK. DML received personal fees/honoraria for a lecture from Biotest UK, for an educational video from Gilead and for a 'round table' discussion with Merck. SP is co-investigator on the grant provided by UKRI/NIHR, Grant number NIHR135366. DR has received consulting fees from OMASS therapeutics. DR has a leadership/fiduciary role in the Heal-COVID trial TMG. BRS reports grant money paid to his employer (Berry Consultants, LLC) from The University of Oxford, from the Sponsor's grant from the UKRI/NIHR, per the statistical design and analyses for the PANORAMIC trial. GH reports that the NIHR funded this study. ML reports funding directly from the PANORAMIC trial (NIHR). ML reports being a RAPIS-TEST (NIHR EME) DMC member. JM reports that this is part of his consulting work for Berry Consultants. SK reports research funding from GSK, ViiV Healthcare, Ridgeback Biotherapeutics, Vir and Merck unrelated to this work. SK reports speaker's fees from ViiV Healthcare and participation on ViiV Healthcare, Pfizer advisory boards. SK reports receiving donation of drugs for clinical studies from ViiV Healthcare, Toyama and GSK. JFS reports receiving research grants from MRC (MR/X004724/1), Wellcome, NIHR, DNDi, Gates and MRC (MR/W015560/1) paid to his institution. JFS reports receiving Pharmacometric consultancy fees from Adrenomed Ltd paid to his institution. JFS reports participation on a Data Safety Monitoring board/Advisory Board for GSK Sotrovimab paediatric programme. MIA reports receiving grants from BTRU – GEMS, Janssen – Cartography, Pfizer – Myst, Prenetics, Dunhill Medical Trust, BMA Trust – Kathleen Harper Fund and Antibiotic Research UK – all paid to the institution. MIA reports receiving consultancy fees from Prenetics and OxDx. MIA reports a planned patent for Ramanomics. MIA reports participation on a Data Safety Monitoring board/Advisory Board for Prenetics. MIA has an unpaid leadership/fiduciary role in the E3 Initiative. NPBT reports his current affiliations with RCGP and NIHR TVSM. NPBT reports a payment for a single episode of participation on the MSD advisory board in July 2021, prior to any knowledge or planning of this trial. OvH reports receiving an NIHR Development and Skills Personal Award. OvH reports receiving consulting fees for MINDGAP BV, with the fees paid to Oxford University Innovation Limited. OvH reports unpaid participation on a Data Safety Monitoring board/Advisory Board for The CHILdren with COugh Cluster Randomised Controlled Trial (CHICO). OvH has an unpaid leadership/fiduciary role in the British Society of Antimicrobial Chemotherapy. CB reports full employment with the Nuffield Department of Primary Care Health Sciences. AU reports receiving consulting fees and payment/honoraria from Merck/MSD and Gilead Sciences. RH reports receiving expenses reimbursed for talks on COVID monoclonal antibodies. PL reports support from the NIHR for the grant to do the PANORAMIC study. NF reports receiving consulting fees from Abbott Diagnostics and GSK, a presentation fee from Abbott Diagnostics, and has stocks in Synairgen PLC.

**Patient and public involvement** Patients and/or the public were involved in the design, or conduct, or reporting, or dissemination plans of this research. Refer to the Methods section for further details.

**Patient consent for publication** Not applicable.

**Provenance and peer review** Not commissioned; externally peer reviewed.

**ORCID iDs**
Oghenekome Gbinigie http://orcid.org/0000-0002-2963-4491
Jienchi Dorward http://orcid.org/0000-0001-6072-1430
May Ee Png http://orcid.org/0000-0001-5876-9363
Gail Hayward http://orcid.org/0000-0003-0852-627X
Mark Lown http://orcid.org/0000-0001-8309-568X
Monique I Andersson http://orcid.org/0000-0003-0619-1074
Nick Francis http://orcid.org/0000-0001-8939-7312
Paul Little http://orcid.org/0000-0003-3664-1873
Christopher C Butler http://orcid.org/0000-0002-0102-3453

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
