## [Reviewer comments · BMJ Open]

ARTICLE DETAILS

TITLE (PROVISIONAL)	Platform Adaptive trial of NOvel antiVIRals for eARly treatMent of COVID-19 In the Community (PANORAMIC): protocol for a randomised, controlled, open-label, adaptive platform trial of community novel antiviral treatment of COVID-19 in people at increased risk of more severe disease.
AUTHORS	Gbinigie, Oghenekome; Ogburn, Emma; Allen, Julie; Dorward, Jienchi; Dobson, Melissa; Madden, Tracie-Ann; Yu, Ly-Mee; Lowe, David; Rahman, Najib; Petrou, Stavros; Richards, Duncan; Hood, KerENZA; Patel, Mahendra; Saville, Benjamin; Marion, Joe; Holmes, Jane; Png, May Ee; Hayward, Gail; Lown, Mark; Harris, Victoria; Jani, Bhautesh; Hart, Nigel; Khoo, Saye; Rutter, Heather; Chalk, Jem; Standing, Joseph; Breuer, Judith; Lavallee, Layla; Hadley, Elizabeth; Cureton, Lucy; Benysek, Magdalena; Andersson, Monique; Francis, Nick; Thomas, Nicholas; Evans, Philip; van Hecke, Oliver; Koshkouei, Mona; Coates, Maria; Barrett, Sarah; Bateman, Clare; Davies, Jennifer; Raymundo-Wood, Ivy; Ustianowski, Andrew; Nguyen-Van-Tam, Jonathan; Carson-Stevens, A; Hobbs, Richard; Little, Paul; Butler, Christopher C.

VERSION 1 – REVIEW

REVIEWER	Lai, Chih-Cheng Chi Mei Medical Center, Liouying, Intensive Care Medicine
REVIEW RETURNED	06-Mar-2023
GENERAL COMMENTS	The study is interesting and the protocol is well-written. Thus, I just have one minor suggestions. Part of finding by this study was published as the following, so please cite this article. https://www.thelancet.com/journals/lancet/article/PIIS0140-6736(22)02597-1/fulltext
REVIEWER	Sommer, Isolde Donau-Universitat Krems, Department for Evidence-based Medicine and Evaluatio
REVIEW RETURNED	08-Mar-2023
GENERAL COMMENTS	This is a very thoroughly written protocol. I have only minor comments (see attachment – contact publisher to view attachment).
REVIEWER	Tong, Steven University of Melbourne, Department of Infectious Diseases
REVIEW RETURNED	22-Apr-2023
GENERAL COMMENTS	Overall, this is a fantastic study. I have no major concerns with the manuscript. Most of my comments are to seek clarifications and

	hopefully improve readability. Page 12 – suggest clarification of wording around ‘between two days of symptom onset and randomisation’. Does this mean patients must be symptomatic first and have a positive test at least 2 days later to be eligible? What if a patient was initially asymptomatic when tested, but then developed symptoms – are they eligible? Page 13 & 14 – suggest clarification – do all participants provide verbal consent? If by telephone / video call, is verbal consent sufficient? I think it just needs some re-ordering of the sentences in this section. Probably: in person or phone consent; if in person, then X happens.., if phone consent, then Y happens. Just a note to include in the limitations, that the identification and consent procedures in themselves will create some selection bias. Only those with the capacity / knowledge to navigate the identification and consent procedures will be enrolled. There is potential for underserved / socio-economically deprived / non English speaking populations to be less likely to enrol. And these may be exactly the groups that are at increased risk of the trial endpoints. Page 14 – stratification by age and vaccination status are mentioned. Could the authors provide more details? Vaccination status is no longer a simple Y/N – is this accounted for? How is age stratification done? Page 15 – are only those who consent screened? Or does screening occur first? Are potentially identifying details collected and stored prior to consent (e.g., DOB)? I’m thinking about the GDPR requirements. I understand that there are ISA specific exclusions. But in this core protocol paper, could the authors provide some indications re how higher risk patients are dealt with. E.g., if a patient qualifies otherwise for paxlovid or monoclonals, are they excluded from PANORAMIC? If they receive paxlovid, would they be eligible for molnupiravir vs usual care (which includes paxlovid)? While all the details will be in the adaptive design report, I think it is worthwhile including a summary of the decision criteria in this manuscript to assist reader understanding. Is there a time / epoch variable for the primary analysis variable? While only concurrent randomisations are included in the primary analysis population, during a pandemic, there may continue to be changes in population characteristics and outcomes over time. Please clarify how ‘all cause hospitalisation’ in a primary outcome, but ‘hospitalisations for pre-existing conditions... do not contribute to our primary outcome’? The second of these seems to be rather subjective. I can understand elective surgery. But other pre-existing conditions? Page 22 – ‘common genetic mutations in patients’. Please clarify if this refers to host or viral mutations? Please confirm if nasopharyngeal swabs are self collected? Page 23 – what is CE-IVD?
--	--

	How are the antivirals obtained? Are these purchased by the trial, supplied by the pharmaceutical company etc? Page 28 – have the investigators achieved the aim of a diverse trial population? Can this be commented upon, at least with the 26,000 participants for the molnupiravir trial? Page 29 – could the authors further elaborate on the weakness of the open label design in terms of patient reported outcomes. Is there evidence from influenza trials? How about the molnupiravir results?
--	---

VERSION 1 – AUTHOR RESPONSE

Reviewer 1 – Comment 1: The study is interesting and the protocol is well-written. Thus, I just have one minor suggestions. Part of finding by this study was published as the following, so please cite this article. https://www.thelancet.com/journals/lancet/article/PIIS0140-6736(22)02597-1/fulltext	We thank the reviewer for this positive feedback. We have now added a reference to the findings of the molnupiravir experience. Please see the reference [ref 18] added on page 10, line 253.
Reviewer 2 – Comment 1: Abstract, pg 5– “I suggest adding vaccination as indicated above”	We recruited our participants in the UK, which had relatively early and widespread rollout of COVID-19 vaccinations, with good take-up. Thus, whilst our trial population was largely vaccinated, prior COVID-19 vaccination was not an inclusion criterion.
Reviewer 2 – Comment 2: Intro pg 8 – “What about remdesivir”	Remdesivir requires intravenous injection and is therefore more suited to inpatient rather than outpatient use. Our trial is treating outpatient treatments for COVID-19.
Reviewer 2 – Comment 3: Pg 10 – Please define standard care or provide a reference	We have now added an explanation of usual care and a supporting reference. Please see page 10, lines 241-242.
Reviewer 2 – Comment 4: Pg 12 – eligibility criteria – “What about vaccination and pregnancy?”	As per our response to comment 1, vaccination status is not part of the eligibility criteria. In order to maximise the trial inclusivity, pregnancy is not an exclusion criterion in the master protocol. However, it features as an exclusion criterion within intervention-specific appendices (ISAs), dependent on the intervention in question.
Reviewer 2 – Comment 5: Pg 13 – ISAs – please provide a reference, I cannot find it.	The ISAs can be found in the appendices – please see pages 97 to 123.
Reviewer 2 – Comment 6: Pg 15 – please address who was responsible for randomisation and how allocation concealment was ensured.	Randomisation was performed using a secure, fully validated, and compliant web-based randomisation system embedded within Spinnaker (a data entry system); there is no means of predicting participant allocation prior to randomisation – please see page 15, lines 353 to 355.

Reviewer 2 – Comment 7: Pg 15 – see comment 5 above	The ISAs can be found in the appendices – please see pages 97 to 123.
Reviewer 2 – Comment 8: Pg 19 – what about subgroup analyses?	Subgroup analyses are performed according to age group, baseline comorbidity status, severity of symptoms at baseline, duration of symptoms at baseline, use of an inhaled corticosteroid steroid at randomisation or during 28 days of follow-up, swab positivity status (PCR positive versus Lateral Flow Device positive), vaccination status, and COVID-19 risk category (as per the UK government description). Additional treatment specific subgroup analyses are detailed in Intervention Specific Appendices of the protocol. Details of subgroup analyses can be found in the statistical analysis plan. This information has been added to the manuscript – see page 20, lines 480 to 486.
Reviewer 3 – Comment 1: Overall, this is a fantastic study. I have no major concerns with the manuscript. Most of my comments are to seek clarifications and hopefully improve readability.	We thank the reviewer for this positive feedback.
Reviewer 3 – Comment 2: Page 12 – suggest clarification of wording around 'between two days of symptom onset and randomisation'. Does this mean patients must be symptomatic first and have a positive test at least 2 days later to be eligible? What if a patient was initially asymptomatic when tested, but then developed symptoms – are they eligible?	This has now been clarified as "...up to two days before symptom onset and randomisation." Please see page 13, line 301.
Reviewer 3 – Comment 3: Page 13 & 14 – suggest clarification – do all participants provide verbal consent? If by telephone / video call, is verbal consent sufficient? I think it just needs some re-ordering of the sentences in this section. Probably: in person or phone consent; if in person, then X happens..., if phone consent, then Y happens.	All participants provide verbal +/- written consent to participate, and the consent form is sent to each participant (electronically or hard copy). This has been deemed sufficient by the ethics committee that approved the trial. We have re-worded this section for clarity – please see pages 13 to 14, lines 325 to 335.
Reviewer 3 – Comment 4: Just a note to include in the limitations, that the identification and consent procedures in themselves will create some selection bias. Only those with the capacity / knowledge to navigate the identification and consent procedures will be enrolled. There is potential for underserved / socio-economically deprived / non English speaking populations to be less likely to enrol. And these may be exactly the groups that are at increased risk of the trial endpoints.	We did recruit participants who lacked capacity if they lived in a care home (see page 14, lines 340 to 341). However, we have added the following to the limitations section of the discussion (see page 29, lines 702 to 705): "However, we recognise that recruitment to the trial requires prospective participants to navigate the registration process, which might mean that people from certain groups, such as non-English speaking populations, may be less likely to enrol in the trial."
Reviewer 3 – Comment 5:	We adjusted for vaccination status as a

Page 14 – stratification by age and vaccination status are mentioned. Could the authors provide more details? Vaccination status is no longer a simple Y/N – is this accounted for? How is age stratification done?	binary variable (vaccinated – yes/no). Given that 93% of our study population has received at least three vaccinations, additional granular analysis of the effect of the timing and dose of vaccinations is unlikely to provide clinically useful information and would be difficult to operationalise in routine care. Age stratification was binary (<50 years vs ≥ 50 years). Clarification of this has been added to page 15, lines 354 to 355.
Reviewer 3 – Comment 6: Page 15 – are only those who consent screened? Or does screening occur first? Are potentially identifying details collected and stored prior to consent (e.g., DOB)? I’m thinking about the GDPR requirements.	Participants can complete some initial screening questions (via telephone, online or face-to-face) to get a broad sense of whether they meet the trial’s eligibility criteria. Consent takes place prior to any participant data being stored.
Reviewer 3 – Comment 7: I understand that there are ISA specific exclusions. But in this core protocol paper, could the authors provide some indications re how higher risk patients are dealt with. E.g., if a patient qualifies otherwise for paxlovid or monoclonals, are they excluded from PANORAMIC? If they receive paxlovid, would they be eligible for molnupiravir vs usual care (which includes paxlovid)?	Participants receiving paxlovid or monoclonals as part of their usual care were eligible to additionally eligible to receive molnupiravir. However, a participant who was receiving paxlovid via usual care would not have been eligible to additionally receive paxlovid through the trial. The following has been added to page 16, lines 396 to 400: “Participants receiving a monoclonal antibody infusion or an antiviral agent as part of their usual care were eligible to receive a (different) antiviral through the trial. However, those at highest risk of an adverse outcome were informed that they were eligible for access to antiviral treatment through NHS services.”
Reviewer 3 – Comment 8: While all the details will be in the adaptive design report, I think it is worthwhile including a summary of the decision criteria in this manuscript to assist reader understanding.	The statistical methods section has been amended – see page 20, lines 476 to 478
Reviewer 3 – Comment 9: Is there a time / epoch variable for the primary analysis variable? While only concurrent randomisations are included in the primary analysis population, during a pandemic, there may continue to be changes in population characteristics and outcomes over time.	The primary analysis for a given intervention includes participants randomised to that intervention and concurrent controls. A sensitivity analysis includes participants recruited prior to the introduction of that agent and participants randomised to other interventions. To account for changes in the population and various other factors (e.g. SARS-CoV-2 variants), the model includes parameters to adjust for temporal drift by estimating the primary endpoint in the usual care group across time via Bayesian hierarchical modelling.

Reviewer 3 – Comment 10: Please clarify how ‘all cause hospitalisation’ in a primary outcome, but ‘hospitalisations for pre-existing conditions... do not contribute to our primary outcome’? The second of these seems to be rather subjective. I can understand elective surgery. But other pre-existing conditions?	We did not wish to contaminate the primary outcome with elective hospitalisations that were clearly related to a pre-existing condition e.g. a total knee replacement for known osteoarthritis of the knee. We appreciate that there could be a degree of subjectivity, however, these decisions were made by suitably trained clinicians.
Reviewer 3 – Comment 11: Page 22 – ‘common genetic mutations in patients’. Please clarify if this refers to host or viral mutations? Please confirm if naso-pharyngeal swabs are self collected?	This refers to viral mutations – please the clarification on page 22, line 549. Naso-pharyngeal swabs are self-collected – clarification of this has been provided on page 23, line 559
Reviewer 3 – Comment 12: Page 23 – what is CE-IVD?	CE-IVD indicates compliance with the European In-Vitro Diagnostic Devices Directive. Clarification of this has been added to the manuscript – please see page 23, lines 568 to 569.
Reviewer 3 – Comment 13: How are the antivirals obtained? Are these purchased by the trial, supplied by the pharmaceutical company etc?	The antiviral agents were supplied to the trial by the UK Antiviral Task Force.
Reviewer 3 – Comment 14: Page 28 – have the investigators achieved the aim of a diverse trial population? Can this be commented upon, at least with the 26,000 participants for the molnupiravir trial?	A comment and additional reference pertaining to this have been added to the manuscript – see page 28-29 lines 699 to 702.
Reviewer 3 – Comment 15: Page 29 – could the authors further elaborate on the weakness of the open label design in terms of patient reported outcomes. Is there evidence from influenza trials? How about the molnupiravir results?	We have further elaborated in the discussion – please see page 29 lines 718 to 719. A reference to the findings of the molnupiravir paper has been added to the discussion section – see line 722. Furthermore, we discuss this topic in relation to the findings of the open-label PRINCIPLE trial (see page 30, lines 727 to 729).

VERSION 2 – REVIEW

REVIEWER	Sommer, Isolde Donau-Universitat Krems, Department for Evidence-based Medicine and Evaluation
REVIEW RETURNED	26-Jun-2023

GENERAL COMMENTS	The authors have adequately addressed all comments. I have no further concerns about this protocol.
---

REVIEWER	Tong, Steven University of Melbourne, Department of Infectious Diseases
REVIEW RETURNED	20-Jun-2023

GENERAL COMMENTS	The revised manuscript has appropriately addressed the issues raised in the first round of reviews.
---